# Dialog-to-Action: Conversational Question Answering Over a Large-Scale Knowledge Base

**Daya Guo**[1]*, **Duyu Tang**[2], **Nan Duan**[2], **Ming Zhou**[2], and **Jian Yin**[1]

[1] The School of Data and Computer Science, Sun Yat-sen University.
Guangdong Key Laboratory of Big Data Analysis and Processing, Guangzhou, P.R.China
[2] Microsoft Research Asia, Beijing, China
{guody5@mail2,issjyin@mail}.sysu.edu.cn
{dutang,nanduan,mingzhou}@microsoft.com

## Abstract

We present an approach to map utterances in conversation to logical forms, which will be executed on a large-scale knowledge base. To handle enormous ellipsis phenomena in conversation, we introduce dialog memory management to manipulate historical entities, predicates, and logical forms when inferring the logical form of current utterances. Dialog memory management is embodied in a generative model, in which a logical form is interpreted in a top-down manner following a small and flexible grammar. We learn the model from denotations without explicit annotation of logical forms, and evaluate it on a large-scale dataset consisting of 200K dialogs over 12.8M entities. Results verify the benefits of modeling dialog memory, and show that our semantic parsing-based approach outperforms a memory network based encoder-decoder model by a huge margin.

## 1 Introduction

We consider the problem of mapping conversational natural language questions to formal representations (e.g., logical form) of their underlying meanings, which would be executed to produce the answer (denotation) [1–7]. We study the problem in a realistic setting that (1) only denotations are available for model training while the underlying logical forms remain unknown, and (2) logical forms will be executed on a large-scale knowledge base (KB) consisting of tens of millions of entities. We believe that KB-based conversational question answering plays an important role in both search engines and intelligent personal assistants (e.g., Siri, Alexa, Cortana/Xiaoice, and Google Now) [8] to improve the ability of multi-turn question answering.

The major challenge of this task is how to interpret the meaning of an utterance in interaction where ellipsis phenomena are frequently encountered. Let's consider the example in Figure 1. The ellipsis of the entity *"he"* in *Q2* refers to *"President of the United States"* in *Q1*. The ellipsis of the entity *"it"* in *Q3* means the answer *R2*. In *Q4*, the ellipsis of the predicate (*"yearEstablished"*) comes from *Q3*. We see that understanding the meaning of conversational utterances requires a good understanding of dialog history. Another challenge is how to efficiently learn the semantic parser from denotations. Online learning by searching legitimate logical forms requires repeated execution on a large-scale knowledge base, which is extremely time-consuming and intolerable.

In this work, we regard the generation of a logical form as the prediction of a sequence of actions [9, 10, 6, 11–16], each of which corresponds to a derivation rule in a simple and flexible grammar. We introduce a generative model that interprets the logical form of an utterance in a top-down manner. A grammar-guided decoder is developed to generate possible action sequences following the grammar.

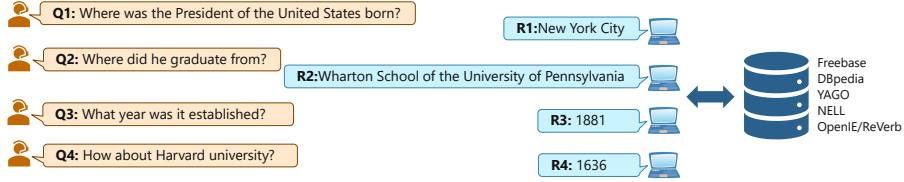

Figure 1: An example illustrating the task of conversational question answering.

To cope with ellipsis phenomena in conversation, we introduce a dialog memory management component that leverages historical entities, predicates, and action subsequences when generating the logical form for a current utterance. To avoid the time-consuming procedure of repeatedly executing on a large-scale knowledge base during training [6], we conduct a breadth-first-search algorithm in advance to obtain pairs of utterances and their action sequences that lead to correct answers. The model is learned by maximizing the likelihood of generating the expected action sequences [5, 11].

We conduct experiments on a large-scale dataset [17] for conversation question answering, which consists of 200K dialogs with 1.6M turns over 12.8M entities from Wikidata. Compared to a memory network enhanced encoder-decoder method [17], our semantic parsing-based approach achieves better performance. We show the benefits of using dialog memory, and observe that our approach performs well on those questions which rely on dialog contexts for resolving ellipsis phenomena.

## 2 Problem Statement

Our goal is to answer questions (utterances) in conversations based on a large-scale open-domain knowledge base (KB). We tackle the problem in a semantic parsing manner that first maps the question into executable logical forms, and then executes the generated logical form on a KB to produce the answer. We would like to learn the semantic parser from denotations, having no luxury of access to the annotated logical form for each utterance. Formally, let $I$ be an interaction consisting of $n$ utterances/questions $\{q_1, q_2, ..., q_n\}^2$. During training, each question $q_i$ is paired with the correct answer $a_i$, without explicit annotation of the correct logical form $z_i$. In the inference process, a logical form $z_i'$ is derived based on the current question $q_i$ and its preceding questions $\{q_1, q_2, .., q_{i-1}\}$. Executing $z_i'$ on knowledge base $K$ produces the outcome $a_i'$.

## 3 Grammar

In this section, we describe the actions we define in this work for generating logical forms. A summary of all the actions are given in Table 1.

Table 1: The base actions we use in this work for generating logical forms.

| Action | Operation | Note |
|---|---|---|
| A1-A3 | $start \rightarrow set|num|bool$ | |
| A4 | $set \rightarrow find(set, r)$ | set of entities with a $r$ edge to $e$ |
| A5 | $num \rightarrow count(set)$ | total number of $set$ |
| A6 | $bool \rightarrow in(e, set)$ | whether $e$ is in $set$ |
| A7 | $set \rightarrow union(set_1, set_2)$ | union of $set_1$ and $set_2$ |
| A8 | $set \rightarrow inter(set_1, set_2)$ | intersection of $set_1$ and $set_2$ |
| A9 | $set \rightarrow diff(set_1, set_2)$ | instances included in $set_1$ but not included in $set_2$ |
| A10 | $set \rightarrow larger(set, r, num)$ | subset of $set$ linking to more than $num$ entities with relation $r$ |
| A11 | $set \rightarrow less(set, r, num)$ | subset of $set$ linking to less than $num$ with relation $r$ |
| A12 | $set \rightarrow equal(set, r, num)$ | subset of $set$ linking to $num$ entities with relation $r$ |
| A13 | $set \rightarrow argmax(set, r)$ | subset of $set$ linking to most entities with relation $r$ |
| A14 | $set \rightarrow argmin(set, r)$ | subset of $set$ linking to least entities with relation $r$ |
| A15 | $set \rightarrow \{e\}$ | |
| A16-A18 | $e|r|num \rightarrow constant$ | instantiation for entity $e$, predicate $r$ or number $num$ |
| A19-A21 | $set|num|bool \rightarrow action_{i-1}$ | replicate previous action subsequence (w/o or w/ instantiation) |

Analogous to the meaning representation of [9], each action in this work consists of three components: a semantic category, a function symbol which might be omitted, and a list of arguments. An argument can be a semantic category, a constant, or an action subsequence. Take *A5* for example: it has a semantic category $num$, a function symbol $count$, and a semantic category $set$ as the only argument. We add *A16-A18* to instantiate entity $e$, predicate $r$, and number $num$, respectively. We add *A19-A21* for replicating a subsequence of previously predicated action sequence from the dialog memory. The derivation of a logical form starts from the semantic category $start$. As derivation processes, the model recursively rewrites the leftmost nonterminal (i.e semantic category) in the logical form by applying a legitimate action. The parsing process terminates until no nonterminals remain.

## 4 Dialog-to-Action

We describe our semantic parsing-based model in this section. Based on sequence-to-sequence learning [18, 19], the model takes a question and its context from interaction history as the input and generates an action sequence. We develop a grammar-guided decoder to control the generation of an action sequence, and a dialog memory management component to leverage historical contexts.

### 4.1 Encoder

Figure 2 illustrates an overview of the proposed model. Since previous questions and answers/responses in conversation are useful contexts, we concatenate them with the current question as an input $x = (x_1, ..., x_T)$. A bidirectional RNN with a gated recurrent unit (GRU) [20] is used as the encoder to convert the input to a sequence of context vector. The forward RNN reads the input in left-to-right direction, obtaining hidden states $(\overrightarrow{h_1}, ..., \overrightarrow{h_T})$. The backward RNN reads reversely and outputs $(\overleftarrow{h_1}, ..., \overleftarrow{h_T})$. We then get the final representation $(h_1, ..., h_T)$ for each word in the source sequence, where $h_j = [\overrightarrow{h_j}; \overleftarrow{h_j}]$. The representation of the source sequence $h_x = ([\overrightarrow{h_T}; \overleftarrow{h_1}])$ is used as initial hidden state of the decoder.

### 4.2 Grammar-guided Decoder

We use GRU with an attention mechanism as a decoder, which generates an action sequence $a_1, ..., a_n$ in a sequential way. As we can see from Figure 2, the decoder parses the dialog to an action sequence, which corresponds to the parsing tree shown in the lower right side. At each time-step $t$, we apply an attention mechanism to obtain the context vector $c_t$ that is computed in the same way as [21]. The concatenation of the context vector $c_t$, the last hidden state $s_{t-1}^{dec}$ and the embedding $v_{t-1}$ of previously predicted action is fed to the decoder to get the current hidden state $s_t^{dec} = GRU(s_{t-1}^{dec}, v_{t-1}, c_t)$. If the previously predicted action is an instantiated action, the embedding $v_{t-1}$ is the representation of the selected constant. The current hidden state $s_t^{dec}$ is used with the same attention mechanism over the inputs to get the context vector $s_t^c$. We then concatenate $s_t^{dec}$ and $s_t^c$ to get final hidden states $s_t$. In order to generate a valid logical form, we incorporate an action-constrained grammar to filter illegal actions. An action is legitimate if its left-hand semantic category is the same as the leftmost nonterminal in the partial logical form parsed so far. We denote the set of legitimate actions at the time step $t$ as $A_t = \{a_1, ..., a_N\}$. The probability distribution over the set is calculated as Equation 1, where $v_i$ is the one-hot indicator vector for $a_i$, $W_a$ is model parameter, and $a_{<t}$ stands for the preceding actions of the $t$-th time step.

$$p(a_i|a_{<t}, x) = \frac{exp(v_i^T W_a s_t)}{\sum_{a_j \in A_t} exp(v_j^T W_a s_t)} \qquad (1)$$

### 4.3 Dialog Memory

Interaction history is very important to generate the logical form of the following utterance. Therefore, we incorporate a dialog memory to maintain information from interaction history. As illustrated in Figure 2, the dialog memory includes three types of information, including entities, predicates, and action subsequences. We describe these aspects one after another.

**Entity** We consider two types of entities from interaction history, coming from the previous question utterance and the previous answer, respectively. The first type is suitable for a common co-reference

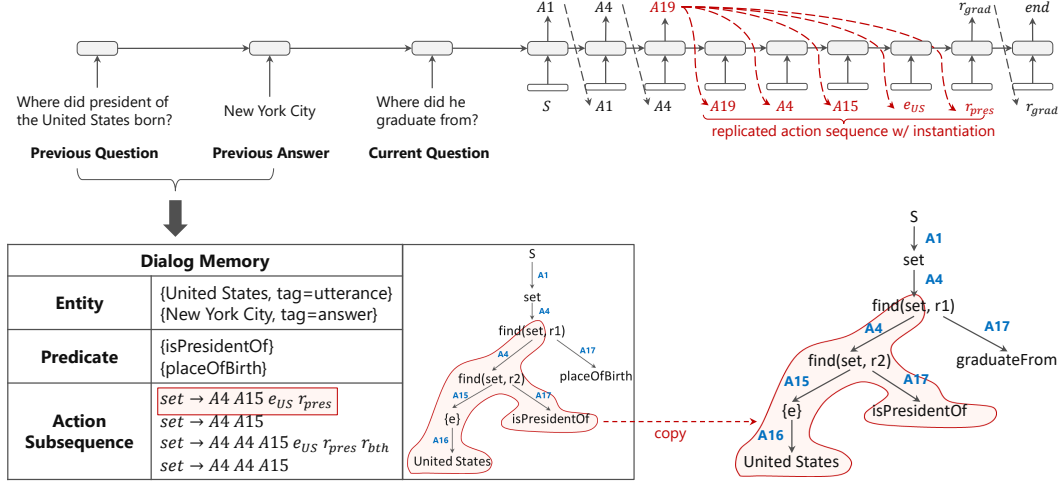

Figure 2: An illustration of the proposed approach. Our approach is the encoder-decoder structure with dialog memory management component. The lower right side is a parsing tree corresponding to the action sequence generated by the decoder.

case where the ellipsis entity comes from the previous utterance, such as "*Q1*: *Where was the President of the United States born*", "*Q2*: *Where did he graduate from*". The second type is suitable for the case in ellipsis entities comes from the previous answer, such as *Q3* and *R2* in Figure 1.

**Predicate** We record the predicates of the previous utterance. This is useful for the scenario where the ellipsis of the predicate occurs. Let us take *Q3* and *Q4* from Figure 1 as an example. The predicate "*yearEstablished*" is not explicitly expressed in *Q4*, yet mentioned in *Q3*.

**Action Subsequences** An action subsequence could be roughly categorized as instantiated or not. Indeed, an action subsequence with instantiation stands for a full or a partial logical form. For example, the first action subsequence in the dialog memory of Figure 2 is identical to the logical form $find(UnitedStates, isPresidentOf)$, which means the president of the United States. The ellipsis of the entity *"he"* in the current question *"Where did he graduate from?"* actually refers to the president of the United States. Therefore, the model executes an action (i.e. *A19*) to replicate the first action subsequence. An action subsequence without instantiation conveys the soft pattern of a logical form. For example, the current question *"And how about China?"* has the same soft pattern as the previous question, but the country mentioned in the previous question should be replaced by *"China"*.

For more details on establishment of the dialog memory, see Appendix A.

## 4.4 Incorporating Dialog Memory

In this section, we present our strategy to replicate contents from dialog memory as decoding processes. This has an influence on *A16-A21*, which we would list as follows.

**Instantiation:** We allow instantiated actions (i.e. *A16-A18*) to access to the dialog memory when the decoder instantiates an entity, predicate, or number. Taking entities as an example. Each entity is assigned one of three tags: previous question, previous answer, or current question. The probability of an entity $e_t$ being instantiated at time-step $t$ is calculated as Equation 2, where $p_g(\cdot)$ is the probability of the tag $g_t$ to be chosen, and $p_e(\cdot)$ is the probability distribution over entities for each tag.

$$p(e_t|a_{<t}, x) = p_e(e_t|g_t, a_{<t}, x)p_g(g_t|a_{<t}, x) \tag{2}$$

The probability distribution of entities $p_e(\cdot)$ is calculated as Equation 3, where $v_e$ is the embedding of entity $e_t$, $W_e$ is model parameter, and $E_{g_t}$ is the set of entities having tag $g_t$. The probability $p_g(\cdot)$ is implemented by a linear layer followed by a softmax function, and the input is $s_t$. The instantiations of predicates and numbers are similar to entities, except that predicates have two kinds of tags (i.e.

previous question and current question) and numbers have only one tag (i.e current question).

$$p_e(e_t|g_t, a_{<t}, x) = \frac{exp(v_e^T tanh(W_e s_t))}{\sum_{e' \in E_{g_t}} exp(v_{e'}^T tanh(W_e s_t))} \quad (3)$$

**Replication:** The model learns to copy a previous action subsequence through choosing *A19-A21*. It has two modes that replicate instantiated or non-instantiated action subsequences. Figures 2 illustrates how the model replicates instantiated action subsequences. In order to obtain instantiated action subsequences of the previous question, we parse the whole previous logical form to a tree and enumerate all subtrees, each of which corresponds to an instantiated action subsequence. Another mode will be described in appendix B. In our model, the probability of a subsequence to be copied is calculated as Equation 4, where $p_m(\cdot)$ is the probability of the mode $m_t$ to be chosen, and $p_s(\cdot)$ is the probability distribution over subsequences for each mode.

$$p(sub_t|a_{<t}, x) = p_s(sub_t|m_t, a_{<t}, x)p_m(m_t|a_{<t}, x) \quad (4)$$

The probability of copying the subsequence $sub_t$, namely $p_s(sub_t|m_t, a_{<t}, x)$, is calculated as follows, where $v_{sub}$ is the representation of $sub_t$, and $E_{m_t}$ is the set of subsequences given mode $m_t$. $v_{sub}$ is obtained by encoding $sub_t$ using a GRU. The calculation of $p_m(\cdot)$ is analogous to $p_g(\cdot)$.

$$p_s(sub_t|m_t, a_{<t}, x) = \frac{exp(v_{sub}^T tanh(W_s s_t))}{\sum_{s_i \in E_{m_t}} exp(v_{si}^T tanh(W_s s_t))} \quad (5)$$

After replicating a subsequence action, the decoder clamps the generation of subsequence length by continuously feeding the subsequence actions one by one. In the inference process, we obtain action subsequences from the predicted logical form with the highest score. Error propagation might occur when the model replicates an incorrect previous logical form, which hurts performance. Therefore, we consider the score of action subsequence as a degree of confidence, which is calculated in the same way as the probability of action subsequences without replication.

## 5    Learning and Inference

At the training phase, instances from training data are labeled with answers while action sequences remain unknown. In order to train our model, we first generate action sequences for each example, and then use an approximate marginal log-likelihood as the objective function. We use a breadth-first-search algorithm from root to generate a set of action sequences $S_a$ that are executed to the correct answer. To cover the replication of action subsequences from dialog memory, we regard action subsequences in $S_a$ which appear in the dialog memory as replicated action subsequences. In order to guarantee the quality of training instances with replication actions, we have a constraint that at least one instantiated constant should be the same. The objective function is the sum of log probabilities of actions, instantiations, and replications, where $\delta(ins, a_t)$ is 1 if $a_t$ is an instantiation action otherwise 0, and $\delta(rep, a_t)$ is the same as $\delta(ins, a_t)$, where $rep$ means a replication action.

$$loss = -\sum_t logp(a_t|a_{<t}, x) - \sum_t \delta(ins, a_t)logp(e_t|a_{<t}, x) - \sum_t \delta(rep, a_t)logp(sub_t|a_{<t}, x)$$
$$(6)$$

We use beam search at the inference phase. For more details on the training and inference procedures used in the experiments, see Appendix C.

## 6    Experiment

We conduct the experiment on the CSQA dataset[3]. The dataset is created based on Wikidata[4], including 152K dialogs for training, and 16K/28K dialogs for development/testing. Questions in dialogs are classified as kinds of types, examples of which are shown in Figure 3. We use the same evaluation metrics employed in [17]. Precision and recall are used as evaluation metrics for questions whose answers are entities, which measures the percentage of correct entities in the output and the percentage of correct entities that are retrieved, respectively. Accuracy is used to measure the performance for questions which produce boolean and numerical answers.

Table 2: Performance of different approaches on the CSQA dataset.

| Methods | HRED+KVmem | | ContxIndp-SP | | D2A | |
|---|---|---|---|---|---|---|
| Question Type | Recall | Precision | Recall | Precision | Recall | Precision |
| Overall | 18.40% | 6.30% | 42.18% | 40.88% | 64.04% | 61.76% |
| Simple Question (Direct) | 33.30% | 8.58% | 94.04% | 88.32% | 93.67% | 89.26% |
| Simple Question (Co-referenced) | 12.67% | 5.09% | 40.29% | 38.55% | 71.31% | 68.41% |
| Simple Question (Ellipsis) | 17.30% | 6.98% | 14.09% | 13.28% | 86.58% | 77.85% |
| Logical Reasoning (All) | 15.11% | 5.75% | 36.23% | 35.91% | 42.49% | 44.82% |
| Quantitative Reasoning (All) | 0.91% | 1.01% | 43.75% | 49.91% | 48.59% | 52.03% |
| Comparative Reasoning (All) | 2.11% | 4.97% | 41.49% | 38.91% | 44.73% | 43.69% |
| Clarification | 25.09% | 12.13% | 0.01% | 0.01% | 19.36% | 17.36% |
| Question Type | Accuracy | | Accuracy | | Accuracy | |
| Verification (Boolean) | 21.04% | | 20.38% | | 45.05% | |
| Quantitative Reasoning (Count) | 12.13% | | 30.60% | | 40.94% | |
| Comparative Reasoning (Count) | 8.67% | | 15.54% | | 17.78% | |

## 6.1 Model Comparisons

Table 2 shows the results of different methods on CSQA data. **HRED+KVmem** [17] is a sequence-to-sequence learning method, which uses a hierarchical encoder and a key-value memory network [22] to compute the representation for the question and its contexts, and then uses an RNN as the decoder to directly produce answers. To demonstrate the effectiveness of dialog memory, we implement a context-independent semantic parser **ContxIndp-SP**, in which the dialog memory is totally removed from the full Dialog-to-Action model. Our full model is abbreviated as **D2A** (short for Dialog-to-Action).

Our approach is a semantic parsing based method, which explicitly manipulates the actions/functions and lets the Seq2Seq model learn how these actions are used to derive the logical form of the question. It could naturally leverage parsed results of previous turn including entities, predicates and action subsequences to handle various ellipsis phenomena. **HRED+KVmem** is a text generation based approach that puts the entire burden of doing reasoning and compositionality to the Seq2Seq model, which struggles at handling all these problems in an implicit way. Results demonstrate that namely semantic parsing approach is more effective to handle complex questions, including quantitative, comparative and logical reasoning. We can also see that incorporating the dialog memory brings significant improvements in co-referenced and ellipsis categories. The results also show that the dialog memory is very important to handle ellipsis phenomena in conversation.

## 6.2 Model Analysis

We conduct ablation analysis to better understand how various components in the dialog memory impact overall performance. We remove entity memory (EM), predicate memory (PM) and action subsequence memory (AM), respectively, to analyze their contribution.

Table 3: Performance of different approaches on the CSQA dataset. EM, PM and AM stand for entities, predicates, and subsequent action sequences from dialog memory, respectively.

| Methods | D2A w/o EM | | D2A w/o PM | | D2A w/o AM | |
|---|---|---|---|---|---|---|
| Question Type | Recall | Precision | Recall | Precision | Recall | Precision |
| Overall | 44.93% | 44.13% | 57.52% | 56.20% | 64.02% | 62.85% |
| Simple Question (Direct) | 93.09% | 88.59% | 93.39% | 88.76% | 93.55% | 88.63% |
| Simple Question (Co-referenced) | 37.95% | 36.54% | 70.42% | 67.89% | 73.36% | 72.01% |
| Simple Question (Ellipsis) | 81.82% | 76.69% | 15.35% | 13.73% | 85.96% | 80.44% |
| Logical Reasoning (All) | 40.85% | 42.76% | 38.20% | 42.37% | 38.69% | 40.11% |
| Quantitative Reasoning (All) | 43.87% | 52.16% | 44.18% | 48.30% | 43.57% | 50.89% |
| Comparative Reasoning (All) | 42.47% | 44.74% | 39.40% | 38.58% | 41.95% | 43.65% |
| Clarification | 1.44% | 1.79% | 0.86% | 1.11% | 17.76% | 16.16% |
| Question Type | Accuracy | | Accuracy | | Accuracy | |
| Verification (Boolean) | 18.44% | | 47.92% | | 50.84% | |
| Quantitative Reasoning (Count) | 38.89% | | 34.04% | | 39.14% | |
| Comparative Reasoning (Count) | 16.51% | | 15.38% | | 16.79% | |

Table 3 shows that the recall and precision of co-referenced questions drop from ∼70% to ∼37% when ablating entity memory (D2A w/o EM), which reveals the importance of entity memory in a co-referenced scenario. We can see that the accuracy of verification questions drops from 45.05% to 18.44%, which means this type of question also needs information on entities from history interaction. After removing the predicate memory, the model (D2A w/o PM) performs poorly in ellipsis questions, dropping from ∼80% to ∼15%. This is consistent with our intuition that the predicate of an ellipsis question comes from the previous question. Results show that removing action subsequence memory (D2A w/o AM) hurts the performance on complex questions including logical reasoning and quantitative reasoning. After analyzing examples of these two types, we observe that ellipsis and co-reference phenomena occur in complex questions, the understanding of which needs to copy complex logical form from previous questions.

To better understand the ability of our semantic parser, we show examples to illustrate the parsing results by our approach (D2A) in Figure 3. As shown, our parser is capable of parsing various types of questions. The 2nd and 3rd examples show that the dialog memory helps the parser replicate entity and predicate from history interaction. Furthermore, replication actions work well in complex questions such as 8th and 9th examples, where previous un-instantiated action subsequences are replicated and instantiation follows.

| id | question type | current question + previous turn | predicted logical form |
|---|---|---|---|
| 1 | Simple Question (Direct) | **Q1**: N/A  **R1**: N/A<br>**Q2**: Who was the dad of Jorgen Ottesen Brahe? | $find(\{Jorgen\ Ottesen\ Brahe\}, father)$ |
| 2 | Simple Question (Coreferenced) | **Q1**: Who was the dad of Jorgen Ottesen Brahe?<br>**R1**: Otte Brahe<br>**Q2**: Who is the spouse of that one? | $find(\{Otte\ Brehe\}, spouse)$ |
| 3 | Simple Question (Ellipsis) | **Q1**: What is the profession of Mkihail Beliaiev?<br>**R1**: Military personnel<br>**Q2**: And also tell me about Brett MacLean | $find(\{Brett\ MacLean\}, occupation)$ |
| 4 | Logical Reasoning (All) | **Q1**: N/A  **R1**: N/A<br>**Q2**: Which administrative territories have diplomatic relations with Italy and are not Derikha present in? | $and(\ diff(\ find(\{Italy\}, reverse(diplomatic\ relation)),$<br>$find(\{Derikha\}, country)),$<br>$find(\{administrative\ territories\}, isA)\ )$ |
| 5 | Quantitative Reasoning | **Q1**: N/A  **R1**: N/A<br>**Q2**: Which works did min number of people do the dubbing for? | $argmin(find(\{voice\ actor\}, isa), reverse(work)\ )$ |
| 6 | Comparative Reasoning | **Q1**: N/A  **R1**: N/A<br>**Q2**: Which musical instruments are played by more number of people than electronic keyboard? | $larger(find(\{musical\ instruments\}, isA)\ ,$<br>$reverse(instrument), count(and($<br>$find(\{electronic\ keyboard\}, reverse(instrument)),$<br>$find(\{people\}, isA))\ ))$ |
| 7 | Verification (Boolean) | **Q1**: N/A  **R1**: N/A<br>**Q2**: Is Arizona Coyotes present in United States of America? | $in(Arizona\ Coyotes\ ,$<br>$find(\{United\ States\ of\ America\}, reverse(country)\ ))$ |
| 8 | Quantitative Reasoning (Count) | **Q1**: How many people have birthplace at Provence?<br>**R1**: 15<br>**Q2**: And how about Peterborough? | $copy(\ count($<br>$find(\{Peterborough\}, reverse(place\ of\ birth)\ )))$ |
| 9 | Comparative Reasoning (Count) | **Q1**: How many musical instruments are played by greater number of people than Body percussion ?<br>**R1**: 30<br>**Q2**: And also tell me about timpani? | $copy(count(\ larger(\ find(\{musical\ instrument\}, isA),$<br>$reverse(instrument)\ \ ,$<br>$count(\ find(\ \{timpani\}, reverse(instrument))\ )\ )\ ))$ |

Figure 3: Examples of the parsing results of D2A. *Q1*, *R1* and *Q2* stand for previous utterance, previous answer and current question, respectively; $copy()$ stands for one of the action from *A19-A21* that replicates previous action subsequence; $reverse()$ is a specific function that could be applied on any predicate, resulting in doubled predicates.

### 6.3 Discussion

To understand the limitations of our approach and shed light on future directions to make further improvements, we randomly select 100 wrongly predicted instances for each category, and summary four main classes of errors as follows.

**Entity Linking.** A common problem is entity linking error when different entities have exactly the same surface name. Based on a balance between accuracy and latency, we represent an entity based on the words it contains in this work, so that there's no difference in their representation. A potential way to alleviate this problem is to learn better word representations by considering the contexts from a knowledge graph [23, 24].

**Spurious Program.** We collect referenced action sequence in an automatic way based on an assumption that a logical form is correct if it could be executed to the correct answer. However, some

of these logical forms are spurious [5], in the sense that they do not represent the meaning of question but get the correct answer. Filtering rules might be useful to filter out spurious logical forms.

**Error Propagation.** The problem of error propagation occurs because our model learns to replicate previously generated action sequences, which might be incorrect despite we consider the probability of the previous logical form. The problem might be alleviated if we incorporate more signals to measure the correctness of a logical form.

**Unsupported Actions.** There exist examples whose logical forms could be not covered by our grammar. An example is "*How many political and administrative territories have diplomatic relationships with France?*", whose answer is "*3 and 15*". Incorporating more actions might improve the coverage, however, the aim of this paper is not to explore dataset-specific grammar, but to show that a flexible grammar works well and dialog memory helps.

# 7   Related Work

Our task closely relates to two lines of works on content-dependent semantic parsing, categorized by the type of supervision used for model learning.

The first line of work learns a context-dependent semantic parser from fully annotated logical forms. [1] first learn a context-independent CCG parser, and then conduct context-dependent substitution and elaboration. [3] produce logical forms using a set of classification models. [7] propose a sequence-to-sequence model with a copying mechanism to replicate previously generated logical form. The main difference between our task and this line of work is that we learn from denotations with no access to annotated logical forms.

The second line of work learns a model from denotations, which could be the answer [6] or the final world state [4]. [2] jointly learn a weighted CCG parser and execute spatial/instructional language in navigation environments. [4] develop a shift-reduce parser and use model projection to reduce the search space. [5] generate tokens of action, constant, and function with a sequence-to-sequence model, and use meritocratic gradient weights and randomized beam search to alleviate the spurious program problem. [25] mapping context-sequential instructions to actions sequence, and propose a learning algorithm that take advantage of single-step reward observations and immediate expected reward maximization. [6] regard SQL generation as action sequence prediction, and search legitimate action sequences through online learning. A special "*subsequent*" action is defined to replicate the entire SQL query of the previously contiguous utterance. Generated SQL query will be executed on a web table to produce the answer. Similar to [6], our definitions of action and structure constraint depend on the language of the target logical form. Compared to their method that only learns to replicate the entire logical form of previous utterance, our model is more flexible and capable of replicating various information from dialog memory including entities, predicates, and action subsequences (i.e. partial logical forms). Our task differs from this line of work in that our logical forms interact with a large-scale knowledge base, which poses new challenges for model training. There also exist memory or encoder-decoder based methods [17, 26, 27] that directly generate an answer utterance as the output of the decoder. Our semantic parsing-based model is essentially different from them in that deep question understanding is required to produce the explicit logical form of the underlying meaning. Our task differs from the "QA+recommendation dialog" task [28, 29] in that they only ask question in the second turn, the intention of which is about the recommended entity of the first turn.

# 8   Conclusion

We present the Dialog-to-Action, a generative model that converts an utterance in conversation to a logical form, which will be executed on a large-scale knowledge base to produce the answer. The model works in a top-down manner following a small and flexible grammar, in which the generation of a logical form is equivalent to the prediction of a sequence of actions. A dialog memory management is developed and naturally integrated in the model, so that historical entities, predicates, and action subsequences could be selectively replicated. The model is effectively learned from denotations without using annotated logical forms. Results on a large-scale dataset demonstrate the effectiveness of considering the dialog memory, and show that our model performs significantly better than a strong memory network-based encoder-decoder model.

## Acknowledgments

This work is supported by the National Natural Science Foundation of China (61472453, U1401256, U1501252, U1611264, U1711261, U1711262). Thanks to the anonymous reviewers and Junwei Bao for their helpful comments and suggestions.

## Footnotes

*Work done while this author was an intern at Microsoft Research.

[2] We use the terms *utterance* and *question* interchangeably.

[3]https://amritasaha1812.github.io/CSQA/

[4]https://www.wikidata.org

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
