[Supplementary Material]

# Dialog-to-Action: Conversational Question Answering Over a Large-Scale Knowledge Base (Appendix)

**Daya Guo**[1]*, **Duyu Tang**[2], **Nan Duan**[2], **Ming Zhou**[2], and **Jian Yin**[1]

[1] The School of Data and Computer Science, Sun Yat-sen University.
Guangdong Key Laboratory of Big Data Analysis and Processing, Guangzhou, P.R.China
[2] Microsoft Research Asia, Beijing, China
{guody5@mail2,issjyin@mail}.sysu.edu.cn
{dutang,nanduan,mingzhou}@microsoft.com

## A    Establishment of Dialog Memory

In practice, we only have the utterances and don't know the entities, predicates, and action subsequences involved in a conversation. In this section, we will describe how to establish the dialog memory.

We obtain the entities mentioned in an utterance by entity detection and entity linking. We regard entity detection as a sequence labeling problem. The task is to assign each token one of two tags, either $Entity$ or $NotEntity$. We use a linear layer followed by a $softmax$ function to predict the tag. Afterwards, we use a simple fuzzy matching function based on word overlap to link each candidate to an entity from the knowledge base. It is helpful to note that the training data provides candidate entities mentioned in an utterance, which is also used in Saha et al. [1], but does not identify the corresponding string in the utterance. Therefore, we use the mapping between entity names and utterances to annotate each token as either $Entity$ or $NotEntity$ for model training.

In practice, we also need to identify the predicates mentioned in a question. We regard it as ranking problem which aims to find the most relevant predicate from all the candidate predicates. We use a BiGRU to encode the question, and then concatenate the question vector with the vector of each candidate predicate to measure their similarity. In practice, we find that a cross-entropy objective works well and we use that for model training. The dataset provides which predicate is mentioned in an utterance, so that we also use that information as the training data for this module. In the inference phase, we select the top two ranked predicates for each utterance.

## B    Replication without instantiation

In this section, we illustrates how the model replicates an non-instantiated action subsequence. An action subsequence without instantiation conveys the soft pattern of a logical form. It's helpful in the case where logical forms of current and previous questions have same pattern but different constants. Taking the dialog in Figure 1 for example, the current question *"And how about China?"* has the same soft pattern as the previous question, but the entity *"United States"* is replaced by *"China"*. In our grammar, replication without instantiation can be applied to handle this kind of questions. As shown in the decoder of Figure 1, an action (i.e. *A19*) is executed to replicate the last action subsequence {*A4→A4→A15*} in the dialog memory. The last action subsequence is an non-instantiated action subsequence that means a complex question asking about the predicate of an entity, in which the entity is obtained through another operation. During replicating an non-instantiated action subsequence, the

Figure 1: An example of replicating an action subsequent without instantiation.

decoder only replicate non-instantiated actions, while constants will be re-instantiated. In order to obtain non-instantiated action subsequences of the previous question, we parse the whole previous logical form to a tree and enumerate all subtrees without leaves, each of which corresponds to an non-instantiated action subsequence.

## C  Learning and Inference

Our grammar allows us to generate highly compositional logical forms. However, many of them are redundant or meaningless (e.g. logical forms with a $union$ operation on two same set). We apply several methods to prune invalid or redundant logical forms during in the searching progress. Firstly, we filter out partial logical forms that lead to invalid result in the knowledge graph before a complete logical form is interpreted. For example, an action $find(e, r)$ will lead to an invalid result if there are no entities linking $e$ with a relation $r$. Secondly, we filter out the logical forms if all the arguments of an action are the same as each other (e.g. $union(\{Apple\}, \{Apple\})$). Thirdly, in order to shrink search space on the CSQA dataset, we set the maximum number of some actions such as $union$, $argmax$ and $larger$ as 1.

Here we list our training details. The beam size of a breadth-first-search algorithm we use to create the supervised date is 1000. We set the dimension of both encoder and decoder hidden state as 300. Word embedding values are initialized with Glove vectors [2]. Predicate embedding values and model parameters are initialized with uniform distribution. We simply calculate the embedding of an entity by averaging the vectors of words it contains. We use the Adam method [3], and set learning rate as 0.001 and batch size as 32. Due to the limitation of computing resources, we only use 15k training examples for each question type to train our **D2A** model. In all the experiments we use the same development set and perform early stopping.

## Acknowledgments

This work is supported by the National Natural Science Foundation of China (61472453, U1401256, U1501252, U1611264, U1711261, U1711262). Thanks to the anonymous reviewers and Junwei Bao for their helpful comments and suggestions.

## Footnotes

*Work done while this author was an intern at Microsoft Research.