[Reviews · NeurIPS 2018]

Reviewer 1



This paper proposes a sequence-to-sequence model to tackle the conversational knowledge based QA. Unlike existing seq2seq models, they additionally incorporate a conversation memory into the model, which enables the model to copy previous action subsequences. Experimental results show that this model significantly outperforms the KVMem based model. The paper is well written. I have a few concerns about this paper: (1) There is a basic assumption throughout the paper, that is, a logical form could be generated using the grammar. This requires that the logical form could be represented as a tree. However, I do think this assumption always holds given the fact that many meaning representations are represented in more complicated structure. For example, the Abstract Meaning Representation (AMR) is represented in a graph structure. (2) It is very hard to follow in the section 6.2. Many symbols in the equation are not described. For example, s_{t} and a_{

Reviewer 2



This paper proposes a semantic parsing method for dialog-based QA over a large-scale knowledge base. The method significantly outperforms the existing state of the art on CSQA, a recently-released conversational QA dataset. One of the major novelties of this paper is breaking apart the logical forms in the dialog history into smaller subsequences, any of which can be copied over into the logical form for the current question. While I do have some concerns with the method and the writing (detailed below), overall I liked this paper and I think that some of the ideas within it could be useful more broadly for QA researchers. Detailed comments: - I found many parts of the paper to be confusing, requiring multiple reads to fully understand. Section 4, for example, describes a "dialog memory", but it is unclear whether and how any of the subcomponents (e.g., predicates, actions) are represented by the model until much later. The paper could really benefit from a concise summary of the method before Section 4, as currently readers are just thrown into details with no context. - The inclusion of actions in the dialog memory is very interesting! This could benefit models trained on other sequential QA datasets as well and strikes me as the most unique part of this paper. - Q2 in Figure 2 ("Where did he graduate from?") is different than the Q2 referenced in the text ("Where is its capital?"), and the figure appears to contain the logical form for the latter question. - The context here is just the previous question. It would be nice if the authors would describe the challenges associated with scaling the approach to the entire dialog history, and perhaps offer a small-scale analysis of the impact of increasing the context size. - The implementation of the "replication actions" are a bit confusing. From my understanding, the decoder first decides to choose a replication action (e.g., A19), and only then decides what entities/predicates/numbers from the dialog memory to replicate. Was this done for efficiency reasons? ICould all of the "action subsequences" be enumerated and then added as separate question-specific actions? - What is the purpose of the "gates" in the replication actions (Sec 6.2)? What does it mean for a gate to be chosen? Don't you already know beforehand where the entities/etc. in the memory come from? Relatedly, why is there only one gate for numbers? Surely numbers can occur in the previous question and previous response... - It would be nice to have some discussion of how time-intensive the learning process detailed in 6.3 / Appendix B is. As the search space is huge, doing BFS is likely very expensive! - Since the dataset is new and the comparisons are only against baselines in the original dataset paper, it's hard to be overly impressed with the huge performance gains. That said, the different ablated versions of the proposed model make up for this and offer nice insights into the relative importance of the different components in the dialog memory. - Section 7.3 is much appreciated; I'd like to see a bit more in the "Unsupported Actions" section about questions that require reasoning across multiple questions.

Reviewer 3



Update: Thank you authors for your response. 1) My original comments regarding the lack of references is misleading. What I meant was that I think the authors should immediately (e.g. in the first paragraph) frame this work in the context of prior work, say, in semantic parsing. The introduction, as currently written, does not provide good grounding for this paper. 2) I think you should explicitly define what you mean by "consistent" such that the reader understands this claim. 4) I am now able to access the website for the dataset (it was down when I submitted my review). In future papers, please use the citation for the peer reviewed paper instead of the Arxiv version. This is especially important for work as crucial as the task your paper is working on. Thanks for your response. I have increased my score. Original text: Even though the authors do not mention semantic parsing in the introduction, It seems to me that the authors proposed a semantic parser (e.g. one that maps utterances to logical forms). This parser does not use logical forms directly as supervision and rather learns from denotations (http://nlp.stanford.edu/pubs/pasupat2016inferring.pdf). This work aims to resolve coreference resolution in multi-turn semantic parsing by examining conversation history. I think the writing needs more polish. In particular, Figure 1 lacks logical forms, which makes the references in the introduction difficult to follow. Moreover, the authors should just use accepted vocabulary instead of inventing their own (e.g. coreference resolution, though I may have misunderstood the motivation). The notation in section 4 and Figure 2 are not introduced. It is not clear what find(set, r1) means until section 5, despite the fact that this understanding is crucial in comprehension of the proposed method. In section 6.2: - it is not clear to me what a subsequence is - it would help if the authors gave an example - What is v_sub? Is it the GRU encoding of sub? - On line 170, what do you mean by "consistent"? The objective function described in 6.3 seems expensive and intractable for larger problems. Moreover, it seems highly prone to false positives from spurious logical forms. The choice of evaluation is not ideal. This particular task seems to be un-refereed. I was also not able to access it at the time of review (e.g. the website was down), nor was I able to find refereed, related works on this task. I would be more convinced by the results if the authors either compare to previously refereed work or demonstrate the effectiveness of this work on an established, refereed task. Minor feedback: - I would refrain from high subjective adjectives such as "huge margin", "outperforms dramatically", etc. and leave the judgement to the reader.